# Serial Change of Endotoxin Tolerance in a Polymicrobial Sepsis Model

**DOI:** 10.3390/ijms23126581

**Published:** 2022-06-13

**Authors:** Min Ji Lee, Jinkun Bae, Jung Ho Lee, Ye Jin Park, Han A Reum Lee, Sehwan Mun, Yun-seok Kim, Chang June Yune, Tae Nyoung Chung, Kyuseok Kim

**Affiliations:** 1Department of Emergency Medicine, CHA University School of Medicine, Seongnam 13497, Gyeonggi, Korea; minji.lee29@gmail.com (M.J.L.); galen97@chamc.co.kr (J.B.); ning3135@naver.com (J.H.L.); yejin6577@naver.com (Y.J.P.); harlee91@naver.com (H.A.R.L.); bogmi0415@gmail.com (S.M.); loupys@naver.com (Y.-s.K.); june1976@hanmail.net (C.J.Y.); hendrix74@cha.ac.kr (T.N.C.); 2Department of Emergency Medicine, CHA Bundang Medical Center, CHA University, Seongnam 13497, Gyeonggi, Korea

**Keywords:** sepsis, immunosuppression, cytokines, rats

## Abstract

Immune suppression is known to occur during sepsis. Endotoxin tolerance is considered a mechanism of immune suppression in sepsis. However, the timing and serial changes in endotoxin tolerance have not been fully investigated. In this study, we investigated serial changes in endotoxin tolerance in a polymicrobial sepsis model. Herein, we used a rat model of fecal slurry polymicrobial sepsis. After induction of sepsis, endotoxin tolerance of peripheral blood mononuclear cells (PBMCs) and splenocytes was measured at various time points (6 h, 12 h, 24 h, 48 h, 72 h, 5 days, and 7 days), through the measurement of TNF-α production after stimulation with lipopolysaccharide (LPS) in an ex vivo model. At each time point, we checked for plasma tumor necrosis factor (TNF)-α, interleukin (IL)-6, and IL-10 levels. Moreover, we analyzed reactive oxygen species (ROS) as measured by 2′,7′-dichlorodihydrofluorescein, plasma lactate, serum alanine aminotransferase (ALT), and creatinine levels. Nuclear factor (NF)-κB, IL-1 receptor-associated kinase (IRAK)-M, and cleaved caspase 3 levels were measured in the spleen. Endotoxin tolerance, measured by TNF-α production stimulated through LPS in PBMCs and splenocytes, was induced early in the sepsis model, starting from 6 h after sepsis. It reached a nadir at 24 to 48 h after sepsis, and then started to recover. Endotoxin tolerance was more prominent in the severe sepsis model. Plasma cytokines peaked at time points ranging from 6 to 12 h after sepsis. ROS levels peaked at 12 h and then decreased. Lactate, ALT, and serum creatinine levels increased up to 24 to 48 h, and then decreased. Phosphorylated p65 and IRAK-M levels of spleen increased up to 12 to 24 h and then decreased. Apoptosis was prominent 48 h after sepsis, and then recovered. In the rat model of polymicrobial sepsis, endotoxin tolerance occurred earlier and started to recover from 24 to 48 h after sepsis.

## 1. Introduction

Traditionally, systemic inflammatory response has been considered as the main pathophysiology of early sepsis, followed by compensatory anti-inflammatory response syndrome [1]. Therapeutic drugs targeting early hyperinflammation have proven ineffective in clinical trials [2]. However, at present, we know that pro- and anti-inflammatory cytokines are produced concurrently in sepsis, causing simultaneous inflammation and immunosuppression [3,4].

The mechanisms underlying immunosuppression in sepsis include lymphopenia, immature neutrophil phenotype, loss of monocyte inflammatory cytokine production (endotoxin tolerance), increased myeloid-derived suppressor cells (MDSCs), production of anti-inflammatory cytokines, loss of expression of HLA-DR in antigen presenting cells (dendritic cells and macrophages), programmed cell death 1 receptor and ligand in T cells, increasing immunosuppressive T cell phenotypes, such as T helper 2 and T regulatory cells, and apoptosis [2,3,4,5].

Endotoxin tolerance, also known as cellular reprogramming, is defined as a reduced responsiveness to lipopolysaccharide (LPS) challenge following a first encounter with endotoxin [6]. It has been tested in septic patients and proposed as one of the mechanisms underlying immunosuppression in sepsis [7,8,9,10,11]. The proposed mechanisms of endotoxin tolerance are the negative regulatory factor of TLR4 signaling pathway, miRNA, apoptosis, as well as chromatin modification and gene reprogramming of immune cells [12].

However, the timing of initiation and dynamic changes in endotoxin tolerance have not been investigated in clinical or animal models of sepsis. To investigate this, we used a polymicrobial sepsis model. We serially checked the endotoxin tolerance of peripheral blood mononuclear cells (PBMCs) and splenocytes in rats of the polymicrobial sepsis model (Figure 1).

## 2. Results

### 2.1. Complete Blood Cells Profiles 

Complete profiles are presented in Appendix A.

### 2.2. Endotoxin Tolerance

TNF-α response after LPS stimulation in PBMCs was significantly decreased until 48 h after sepsis, approximately 0.1% of the sham group, and then started to recover. It fully recovered on day 5 after sepsis (Figure 2a).

In splenocytes, TNF-α secretion after LPS stimulation was also significantly attenuated until 24 h after sepsis (nearly no response), and then fully recovered on day 7 after sepsis (Figure 2b).

Endotoxin tolerance in PBMCs and splenocytes was more prominent in the severe sepsis model (Figure 2c,d).

### 2.3. Reactive Oxygen Species (ROS)

DCF-DA in plasma peaked at 12 to 24 h, and then decreased (Figure 3a).

### 2.4. Cytokines

Cytokine kinetics are different among cytokines. Plasma TNF-α level (Figure 3b) peaked earlier, followed by IL-6 (Figure 3c) and IL-10 (Figure 3d). IL-10 remained at an elevated level for a longer period of time compared with IL-6 (Figure 3d).

### 2.5. Plasma Lactate, Serum ALT, and Creatinine

Lactate and ALT levels peaked at 24 h after sepsis. The creatinine level peaked 48 h after sepsis, indicating that an acute kidney injury occurred at a later time (Figure 4).

### 2.6. Pathologic Evaluation of Spleen

Lymphoid follicles were reduced from 6 h to 5 days, and significantly fewer lymphoid follicles appeared at 48 h, 72 h, and 5 days (Figure 5a, Appendix A).

### 2.7. Apoptosis of Spleen

The level of cleaved caspase 3 in the spleen increased from 6 to 120 h, with a significant peak at 48 h (Figure 5b, Appendix A), which is consistent with the pattern of lymphoid follicles in spleen.

### 2.8. NF-κB and IRAK-M Levels in Spleen

The phosphorylated p65 in the spleen increased from 6 to 48 h and showed a significant increase at 6 and 48 h (Figure 5c, Appendix A). The phosphorylation level of p65 decreased after 72 h. The abundance of IRAK-M increased 12 to 24 h after sepsis, and then decreased (Figure 5d, Appendix A).

## 3. Discussion

This study is the first to demonstrate serial changes in endotoxin tolerance in a polymicrobial sepsis model. We found that endotoxin tolerance occurs relatively early in sepsis, peaking at 24 to 48 h. It started to recover after 24 to 48 h of sepsis, with full recovery on day 7. Moreover, the magnitude of endotoxin tolerance was associated with the severity of sepsis.

Interestingly, we noted a discrepancy between the levels of plasma cytokines (plasma inflammatory markers) and the capacity of immune cells to respond to stimulation (endotoxin tolerance or immune reservoir in PBMCs and splenocytes). Plasma cytokine levels indicated that there were inflammatory surroundings in the plasma, while immune paralysis was ongoing in PBMCs and splenocytes. 

This discrepancy has major implications for therapeutic strategies for the treatment of sepsis. To date, most anti-inflammatory agents have proven ineffective. Currently, the immune-enhancing treatment is undergoing considerable research and development. However, the co-existence of hyperinflammation and immunosuppression may require another approach.

We found that apoptosis of the spleen occurred and nadired at 48 h after sepsis, and then recovered. Although we did not investigate the causal effect of apoptosis and endotoxin tolerance in this study, considering previous studies on the association between endotoxin tolerance and apoptosis [11], we might infer that apoptosis of immune cells could partly explain endotoxin tolerance, though this requires further investigation.

The immune-compromised status in sepsis was first reported in the mid-1970s [12], and was proposed as a compensatory anti-inflammatory response syndrome [13]. However, compensatory anti-inflammatory response is considered to occur later in sepsis, after pro-inflammatory responses [13]. It was first reported in the mid-2000s that the production of anti- and pro-inflammatory cytokines occurs simultaneously [14]. Other studies have suggested simultaneous induction of innate (both proinflammatory and anti-inflammatory) and suppression of adaptive immune response in both preclinical and clinical studies [15,16,17,18,19,20]. With these studies, the term mixed antagonist response syndrome was suggested. However, no study has yet investigated serial changes in immune suppression in a polymicrobial sepsis model.

IRAK-M is a negative regulator of TLR signaling and regulates innate immune homeostasis [21,22], a characteristic trait of endotoxin tolerance. In this study, IRAK-M showed a peak at 12 to 24 h after sepsis, and then decreased, which occurred a little earlier than the results of endotoxin tolerance.

Moreover, we found serial changes in the pathophysiology of sepsis, of which cytokine changes occur first, followed by ROS and organ damage. This sequence seems reasonable.

Our measurement of serial plasma cytokines indicated differences in their dynamic changes compared with a previous study [14]. In our study, TNF-α peaked at 6 h after sepsis, while another study showed a 24-h peak level of TNF-α. However, this variation could be due to differences in research models and the severity of sepsis. Our results are similar to previously reported clinical data in terms of patterns of IL-6 and IL-10 [23], which showed a peak level at 24 h. Furthermore, we found that lactate, ALT, and creatinine in the blood peak later than cytokines and ROS, which might imply organ injury following inflammatory and oxidant injury.

This study has several limitations. First, although fecal slurry or cecal ligation and puncture models seem to be the best models for mimicking clinical sepsis, gaps in knowledge (between preclinical and clinical sepsis) complicate the translation of our findings to clinical conditions. Second, we could not specify the mechanisms of these changes. We investigated the apoptosis of the spleen and IRAK-M level, which is not a causal relationship, but a simple association. This requires further investigation. Moreover, we should investigate another mechanism concerning endotoxin tolerance with this model. These might include miRNA and chromatin modification and gene reprogramming of immune cells. With these further studies, we could develop newer treatment options. The next generation of treatments could be separate (or opposite) targets with different treatment options, i.e., anti-inflammatory agents to reduce or antagonize increased plasma cytokines, and immune-enhancing/anti-apoptotic drugs to augment the immune function of immunosuppressed immune cells. Third, we could not identify which PBMCs or splenocytes were involved in the induction of endotoxin tolerance. There are monocytes, dendritic cells, macrophages, T and B lymphocytes, and natural killer cells in PBMCs and splenocytes. However, monocytes and macrophages are recognized as immune cells for endotoxin tolerance [24]. Finally, in our model, there is a confounder. This model needs aseptic surgical trauma during median laparotomy. However, this procedure was performed in all animals, which might minimize the confounding effects.

In conclusion, we investigated that endotoxin tolerance occurred in a dynamic manner. Moreover, we found that it occurred earlier in sepsis and recovered within 7 days in this polymicrobial infection model. This model can be used to test new immunomodulatory drugs for sepsis.

## 4. Materials and Methods

### 4.1. In Vivo Sepsis Model Induction

This study was approved by the Institutional Animal Care and Use Committee of the authors’ institute (CHA University IACUC—200078), in accordance with the National Institutes of Health Guidelines. This study was carried out in compliance with the ARRIVE guidelines. Male Sprague-Dawley rats, weighing 270 to 330 g, were used as a research model. The rats, donors, as well as treatment recipients were housed in a controlled environment (room temperature 20~24 °C, humidity 40~60%) with access to standard food and water ad libitum for 7 days before the experiment.

We used a body weight-adjusted polymicrobial intra-abdominal infection sepsis model, in accordance with a previous study [25]. The cecal ligation and puncture model has been widely used in animal study of sepsis, but it has some disadvantages. Therefore, we used the cecal slurry model [26]. In brief, donor rats were anesthetized with an intramuscular injection of Zoletil (50 mg/kg) and xylazine (10 mg/kg). A midline laparotomy was performed, and the cecum was extruded. A 0.5 cm incision was performed in the antimesenteric surface of the cecum and the cecum was squeezed to expel feces. The donor rats were then euthanized. The feces were collected and weighed, then diluted with 5% dextrose saline at a ratio of 1:3. In sepsis induction, fecal-recipient rats were anesthetized as described above. Thereafter, 0.5 cm midline laparotomy was performed and fecal slurry was administered into the peritoneal cavity. The fecal slurry was vortexed to obtain a homogeneous suspension before administration into the intraperitoneal cavity. The volume of fecal slurry administered to each animal was adjusted based on the body weight of the recipient rat. We administered subcutaneous fluid resuscitation (30 mL/kg 5% dextrose saline). In addition, imipenem was injected subcutaneously at a dose of 25 mg/kg twice daily for 2 days. Moreover, we did not use pain killers.

*Experiment 1*: The rats were randomly assigned to the study group at 6 h, 12 h, 24 h, 48 h, 72 h, 5 days, and 7 days. The stratified randomization by weight was performed by an assistant in the experiments. We sacrificed the animals at each allocated time points. If the animals could not survive to the allocated time points, they were excluded from the analysis. During the observation period, an employee of the animal research center monitored the animals twice per day. If the animal was doomed to die, the research team would take notice and make the decision for euthanasia.

*Experiment 2*: We also investigated the effect of sepsis severity on endotoxin tolerance. For this, we induced mild and severe sepsis models with different doses of fecal slurry. In our previous experiments, the mortality rate of mild and severe sepsis models was 0 and 50%, respectively. We tested endotoxin tolerance 24 h after septic insults.

In all of the experiments, 198 animals were used. Furthermore, we did not hypothesize that confounders could affect the results with this study design, thus it was not controlled.

### 4.2. Complete Blood Cells Profiles

Complete blood count was performed with an Hemavet 950 (Drew Scientific, Waterbury, CT, USA).

### 4.3. Measuring Endotoxin Tolerance through Ex Vivo PBMCs and Splenocyte Stimulation with LPS

PBMCs and splenocytes were isolated at each time point after sepsis induction. PBMCs were isolated using the Ficoll gradient method [27]. Isolated PBMCs were stimulated with LPS to observe and compare the levels of immune paralysis. Tumor necrosis factor (TNF)-α levels were measured 5 h after LPS stimulation. Isolated PBMCs were seeded at a density of 1 × 10^5^ cells/mL in 96-well plates, and 100 ng/mL LPS (Escherichia coli O111: B4, Sigma-Aldrich, St. Louis, MO, USA) was added to each well. After 5 h, the culture medium was collected, and TNF-α levels were analyzed using a TNF-α enzyme-linked immunosorbent assay (ELISA) kit (ab236712, Abcam, Cambridge, MA, USA).

Splenocytes were isolated and stimulated with LPS to compare immune paralysis [28]. TNF-α levels were measured 5 h after LPS stimulation. Isolated splenocytes were seeded at a density of 5 × 10^5^ cells/mL in 6-well plates, and 1 μg/mL LPS (Escherichia coli O111: B4, Sigma-Aldrich, St. Louis, MO, USA) was added to each well. After 5 h, the culture medium was collected, and TNF-α was analyzed using a TNF-α ELISA kit (R&D Systems, Inc., Minneapolis, MN, USA).

### 4.4. Lactate Assay

Lactate concentrations in the plasma were evaluated using a lactate colorimetric assay kit (BioVision, Milpitas, CA, USA). The sample and reaction mix buffer were added to the wells and incubated for 30 min at room temperature. After 30 min, the optical density of each well was measured at 450 nm using a VersaMax microplate reader (SoftMax Pro software, Molecular Devices, San Jose, CA, USA).

### 4.5. Serum Alanine Aminotransferase (ALT) and Creatinine

Serum ALT levels were measured using an ALT activity colorimetric/fluorometric assay kit (#K752-100, BioVison, CA, USA). The optical density at 570 nm was measured using a VersaMax microplate reader (SoftMax Pro software, Molecular Devices, CA, USA).

Serum creatinine levels were measured using a creatinine colorimetric/fluorometric assay kit (#K625-100, BioVison, CA, USA). The optical density at 570 nm was measured using a VersaMax microplate reader (SoftMax Pro software, Molecular Devices, San Jose, CA, USA).

### 4.6. Cytokine Measurements

The levels of the cytokines interleukin (IL)-6 (R6000B, R&D Systems, Inc., Minneapolis, MN, USA), IL-10 (ab214566, Abcam, MA, USA), and TNF-α (ab236712, Abcam, MA, USA) in plasma were measured using ELISA kits, in accordance with the manufacturer’s instructions. The optical density at 450 nm was measured using a VersaMax microplate reader (SoftMax Pro software, Molecular Devices, CA, USA).

### 4.7. Assessment of Plasma Reactive Oxygen Species (ROS) in Plasma

Plasma 2′,7′-dichlorodihydrofluorescein (DCF) levels were analyzed, in accordance with the reagent manufacturer’s instructions.

### 4.8. Western Blotting

Spleen tissues were lysed using the Pierce radioimmunoprecipitation assay buffer supplemented with a protease inhibitor and phosphatase inhibitor cocktail (Thermo Fisher Scientific, Waltham, MA, USA). Protein concentrations were measured using a bicinchoninic acid protein assay kit (Thermo Fisher Scientific). A total of 50 µg of protein was separated by 10 to 12% SDS-PAGE gel electrophoresis, and then transferred to a polyvinylidene fluoride membrane (GE Healthcare, Chicago, IL, USA). Membranes were blocked in 5% bovine serum albumin for 1 h at room temperature, then membranes were incubated with the primary antibodies [glyceraldehyde 3-phosphate dehydrogenase (GAPDH, 1:5000, sc-32233, Santa Cruz Biotechnology, Dallas, TX, USA), IL-1 receptor-associated kinase (IRAK)-M (1:1000, ab8116, Abcam), and p-p65 (1:1000, 3033, Cell Signaling Technology, Danvers, MA, USA)], and cleaved caspase 3 (1:1000, #9661S, Cell Signaling Technology) at 4 °C overnight. Membranes were washed with TBST and incubated with horseradish peroxidase-conjugated anti-mouse immunoglobulin G (IgG) (1:1000, GTX213111-01, GeneTex, Irvine, CA, USA) or anti-rabbit IgG (1:1000, GTX213110-01, GeneTex) for 1 h at room temperature. Membranes were washed with TBST, and protein expression was optimized using a Clarity Western ECL substrate (Bio-Rad Laboratories, Hercules, CA, USA). The blots were detected using an LAS-4000 luminescent image analyzer (GE Healthcare, Chicago, IL, USA).

### 4.9. Spleen Pathology Study

Isolated spleens from rats were fixed in 4% paraformaldehyde and embedded in paraffin. Embedded samples were sectioned at a thickness of 5 µm and stained with hematoxylin and eosin (H&E). The slides were scanned with an Axio Scan Z1 (Carl Zeiss Microscopy, White Plains, NY, USA). The number of lymphoid follicles is known to be associated with severe sepsis and apoptosis in the spleen [29,30]. Lymphoid follicle numbers were counted per 10 square millimeters of the spleen using the Zen 3.1 Blue edition program (Carl Zeiss Microscopy) in a blinded manner with three observers.

### 4.10. Effects on Apoptosis in the Spleen

Spleen tissues (50 mg) were homogenized and lysed using the radioimmunoprecipitation assay buffer (RC2002-050-00, Biosesang, Seongnam-si, Korea) with a protease inhibitor and phosphatase inhibitor cocktail (P8340, P5726, Sigma-Aldrich). The isolated proteins were then stored at −70 °C. The cleaved caspase 3 levels were measured by Western blotting using caspase 3 antibody (1:1000, 9662, Cell Signaling Technology) or GAPDH.

### 4.11. Statistical Analysis

The Shapiro−Wilk test was performed to determine the normality of the data. Normally distributed data are presented as mean ± standard deviation and were compared using independent *t*-tests. If the data did not fit a normal distribution, they were presented as the median and interquartile range and were analyzed using the Mann−Whitney U test or Kruskal−Wallis test. Fisher’s exact test was used for categorical variables. The basic principle in multiple comparison is ANOVA or Kruskal−Wallis test with post-hoc analysis. However, we have too many groups to compare (eight groups), thus in some cases, we compare time-specific group only with sham group. This is the first study to investigate the serial change of endotoxin tolerance in polymicrobial sepsis model. Therefore, we could not expect the results, which made the sample size impossible. We enrolled 6–12 animals at each time point as the usual sample size of the animal studies. Statistical significance was defined as *p*-value of <0.05. All of the analyses were performed using R software.

## Figures and Tables

**Figure 1 ijms-23-06581-f001:**
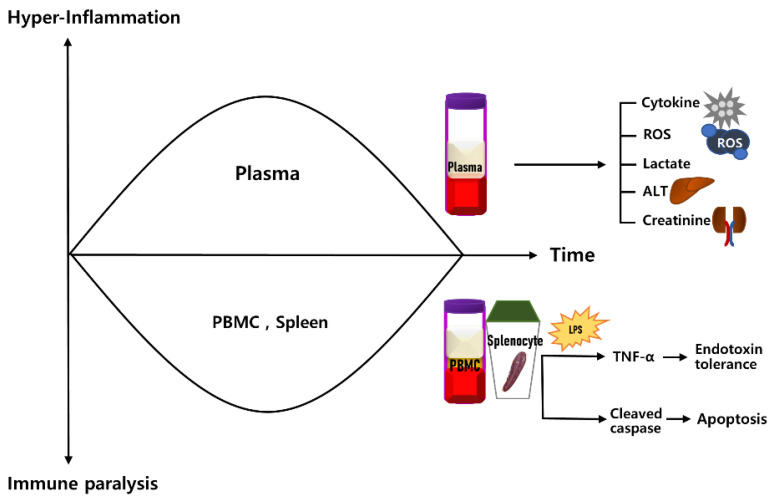
Overview of the serial change of endotoxin tolerance in experimental sepsis model. Schematics showing hyperinflammation in plasma level, but simultaneously, immune suppression and apoptosis in PBMCs and splenocytes.

**Figure 2 ijms-23-06581-f002:**
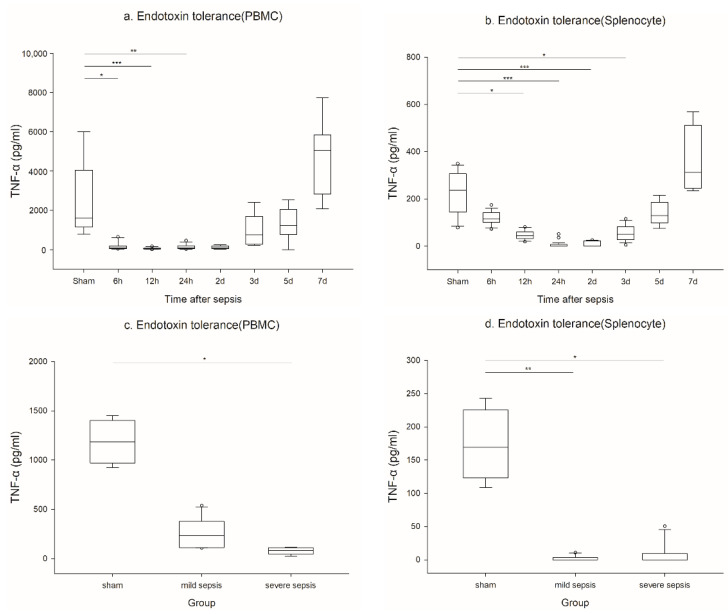
Serial change of endotoxin tolerance in PBMCs and splenocytes: (**a**) Endotoxin tolerance in PBMCs (sham n = 9, 6 and 12 h n = 11, 24 h n = 12, 2 D n = 5, 3, 5, and 7 D n = 6); (**b**) endotoxin tolerance in splenocytes (sham, 2 D n = 11, 6 h n = 18, 12 h n = 12, 24 h n = 29, 3 D n = 16, 5 and 7 D n = 6); (**c**) endotoxin tolerance in PBMCs in mild and severe sepsis models (sham n = 4, mild n = 10, and severe n = 7); (**d**) endotoxin tolerance in splenocytes in mild and severe sepsis models (sham n = 4, mild and severe n = 12). *** *p* < 0.001, ** *p* < 0.01, and * *p* < 0.05 compared with the sham group. PBMCs: Peripheral blood mononuclear cells.

**Figure 3 ijms-23-06581-f003:**
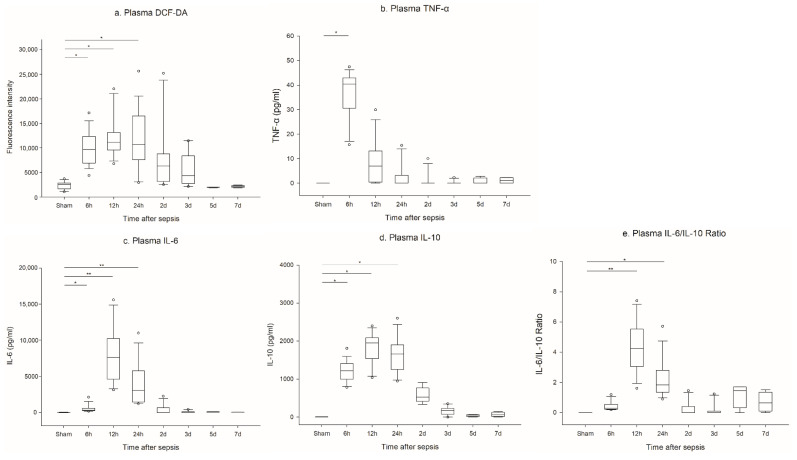
Serial change of plasma reactive oxygen species and cytokines: (**a**) Plasma DCF-DA (sham 2 D n = 11, 6 and 24 h n = 17, 12 h n = 12, 3 D n = 10, 5 and 7 D n = 4); (**b**) plasma TNF-α (sham n = 14, 6 h n = 17, 12 h n = 12, 24 h n = 17, 2 D n = 11, 3 D n = 10, 5 and 7 D n = 4); (**c**) plasma IL-6 (sham n = 9, 6 h n = 18, 12 h n = 12, 24 h n = 15, 2 D n = 11, 3 D n = 10, 5 and 7 D n = 4); (**d**) plasma IL-10 (sham n = 9, 6 h n = 18, 12 h n = 12, 24 h n = 16, 2 D n = 9, 3 D n = 10, 5 and 7 D n = 4); (**e**) plasma IL-6/IL-10 ratio (sham n = 3, 6 h n = 18, 12 h n = 12, 24 h n = 15, 2 D n = 11, 3 D n = 10, 5 and 7 D n = 4). ** *p* < 0.001 and * *p* < 0.05 compared with the sham group.

**Figure 4 ijms-23-06581-f004:**
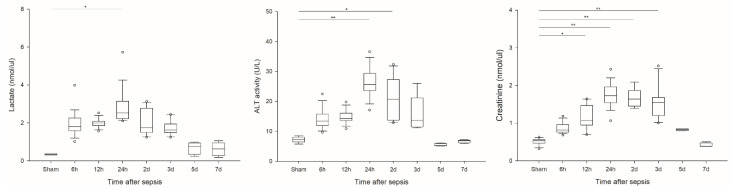
Serial change of plasma lactate, serum ALT, and creatinine: (**a**) Plasma lactate (sham n = 2, 6 h n = 18, 12 h n = 12, 24 h n = 16, 2 D n = 11, 3 D n = 10, 5 and 7 D n = 4); (**b**) serum ALT (sham n = 5, 6 h n = 15, 12 h n = 12, 24 h n = 16, 2 D n = 10, 3 D n = 8, 5 and 7 D n = 4); (**c**) serum creatinine (sham n = 14, 6 h n = 18, 12 h n = 12, 24 h n = 16, 2 D n = 9, 3 D n = 10, 5 and 7 D n = 4). ** *p* < 0.001 and * *p* < 0.05 compared with the sham group. ALT: Alanine aminotransferase.

**Figure 5 ijms-23-06581-f005:**
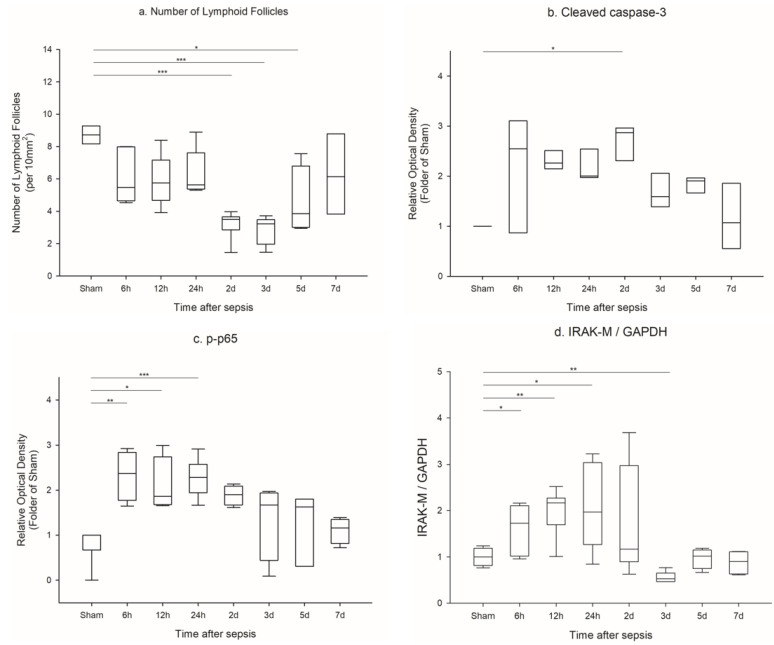
Serial change of pathology of spleen, apoptosis, NF-kB, and IRAK-M: (**a**) Pathologic changes in spleen, lymphoid follicles counted (sham and 7 D n = 3, 6 h, 12 h, and 2 D n = 6, 24 h, 3 D n = 5, and 5 D n = 4); (**b**) cleaved caspase 3 expression in spleen (n = 3 per group); (**c**) phosphorylated p65 in spleen (sham n = 7, 6 h, 12 h, 2 D, 3 D, and 7 D n = 4 per group, 24 h n = 6, and 5 D n = 3); (**d**) IRAK-M in spleen (sham, 5 D, and 7 D, n = 4 per group, 6 h, 24 h, 2 D n = 8 per group, and 12 h, 3 D n = 6). The blots were cropped and full-length blots are presented in Appendix A. The samples were derived from the same experiment. *** *p* < 0.001, ** *p* < 0.01, and * *p* < 0.05 compared with the sham group.

## Data Availability

The datasets generated and analyzed during the current study are available from the corresponding author on reasonable request.

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
