# Peer review of "Serial Change of Endotoxin Tolerance in a Polymicrobial Sepsis Model"

_ijms, 2022, doi:10.3390/ijms23126581_

Round 1

Reviewer 1 Report

Dear Authors,

you presented a nice written work, demonstrating the concurrent presence of hyper-inflammation and immune-paralysis in sepsis, which is a very interesting field, as therapeutic interventions, capable to modify the impact of underlying pathophysiologic mechanisms, may influence the clinical outcome of septic patients. Please pay attention to the following points:

1.      Materials and Methods, Lines 194-196: feces could be extracted from the bowel of the study subjects after median laparotomy and squeezing of cecum. Please explain why you should act so invasively in order to gain the fecal samples for your study. Are there any advantages in comparison to a non-invasive collection of feces?

2.      Materials and Methods, Lines 197-199: you induced an abdominal sepsis by placing fecal samples into the peritoneal cavity of your study subjects. Were those the same study subjects that underwent a median laparotomy in order to gain the fecal samples or completely different? If they were different, please explain how you randomized them and refer if you euthanatize them and the end of the study also.

3.      Materials and Methods, Lines 202-204: please explain why did you initiate a rescue therapy with fluids and antibiotics in your study subjects. Was that in order to prolong their survival until you close your measurements?

4.      Materials and Methods, Lines 212-215: How did you know that your study subjects had an isolated abdominal sepsis without bacteremia? A possible bacteremia could have influenced your results enormously. Did you take any blood cultures? Please discuss this point in your manuscript.

5.      Materials and Methods, Lines 217-218: I believe that there was a confounder in your study and this is the aseptic surgical trauma after median laparotomy. Please discuss this point in your manuscript.

With Best Regards

Author Response

 Reviewer 1

Dear Authors,

you presented a nice written work, demonstrating the concurrent presence of hyper-inflammation and immune-paralysis in sepsis, which is a very interesting field, as therapeutic interventions, capable to modify the impact of underlying pathophysiologic mechanisms, may influence the clinical outcome of septic patients. Please pay attention to the following points:

  1. Materials and Methods, Lines 194-196: feces could be extracted from the bowel of the study subjects after median laparotomy and squeezing of cecum. Please explain why you should act so invasively in order to gain the fecal samples for your study. Are there any advantages in comparison to a non-invasive collection of feces?

-->We appreciate your valuable comments. As you commented, classically cecal ligation and puncture (CLP) model has been used in animal study model of sepsis. However, recently, cecal slurry peritonitis model used in our experiments has risen as a better model. In CLP model, the followings could be disadvantages (as in PMID 31885502 table 1); Uncontrollable because of too many variables; fulminant sepsis vs. survival with intra-abdominal abscesses; no clinical setting of generalized peritonitis). However, the cecal slurry model has not such disadvantages (as in PMID 31885502 table 1). We added this in brief in method section with reference as follows: Classically cecal ligation and puncture model has been widely used in animal study of sepsis, but it has some disadvantages. Therefore, we used cecal slurry model.

  1. Materials and Methods, Lines 197-199: you induced an abdominal sepsis by placing fecal samples into the peritoneal cavity of your study subjects. Were those the same study subjects that underwent a median laparotomy in order to gain the fecal samples or completely different? If they were different, please explain how you randomized them and refer if you euthanatize them and the end of the study also.

-->We apologize for the unclear phrase. In our model, there are donor rats and recipient rats. We collected fecal materials from donor rats, and we introduced it to recipient rats. We revised it more clearly as follows: In sepsis induction, fecal-recipient rats were anesthetized

We did not randomize rats to donor or recipient rats. Instead, we used the most heavier ones as donor because sometimes, there are outlier in body weight even though we used the same age rats. As in method section, we used stratified randomization in sepsis rats (fecal recipient rats) according to the body weight, so we would like to use the rats of even body weight. So, we used most heavier rats as donors. After donation of fecal materials, donor-rats were euthanized. We added this to the method section as follows; The donor rats were then euthanized.  

  1. Materials and Methods, Lines 202-204: please explain why did you initiate a rescue therapy with fluids and antibiotics in your study subjects. Was that in order to prolong their survival until you close your measurements?

--> We appreciate your comments. Rescue fluid therapy and antibiotics were used to mimic clinical scenario as possible as we could. In these days, the rescue fluid therapy and antibiotics were recommended in animal study of sepsis (PMID 30106875).

  1. Materials and Methods, Lines 212-215: How did you know that your study subjects had an isolated abdominal sepsis without bacteremia? A possible bacteremia could have influenced your results enormously. Did you take any blood cultures? Please discuss this point in your manuscript.

-->We appreciate your important comments. We induced sepsis into rats. Sepsis is not localized infection, but systemic inflammation. So, there should be bacteremia in our model. In this study, we did not perform blood cultures, but in previous our studies, bacteria were isolated from blood culture (PMID 35330172, 33801494, 33413559, 27884339).

  1. Materials and Methods, Lines 217-218: I believe that there was a confounder in your study and this is the aseptic surgical trauma after median laparotomy. Please discuss this point in your manuscript.

-->We appreciate your great comments. As you commented, we added the followings in discussion; In our model there is a confounder. This model needs aseptic surgical trauma during median laparotomy. However, this procedure was performed in all animals, which might minimize the confounding effects.

Reviewer 2 Report

Dear Authors,

The manuscript entitled " Serial change of endotoxin tolerance in a polymicrobial sepsis model"  is a very well-structured article. The article describes the serial changes in endotoxin tolerance in a polymicrobial sepsis model. Only minor revisions, i have proposed for the current manuscript.

1) In the introduction section, lines 45-48, i think more details must be added regarding the mechanism of endotoxin tolerance. 

2) The introduction should contain more information regarding the mechanisms that are activating in the sepsis model such as the immunosuppression. Also the responsible cellular populations for the mediated immunosuppresion are required to be described.

3) The results are well described and no changes are required.

4) Discussion section, lines 137-140, The next generation of treatments could...immune cells. I think that this paragraph cannot be supported by the results of this study. It could be as a conclusion. I would suggest to the authors, that this paragraph needs to be removed.

Author Response

  Reviewer 2

Dear Authors,

The manuscript entitled " Serial change of endotoxin tolerance in a polymicrobial sepsis model" is a very well-structured article. The article describes the serial changes in endotoxin tolerance in a polymicrobial sepsis model. Only minor revisions, i have proposed for the current manuscript.

  • In the introduction section, lines 45-48, i think more details must be added regarding the mechanism of endotoxin tolerance. 

-->We appreciate your great comments. We added the details of the mechanisms of endotoxin tolerance as follows; The proposed mechanisms of endotoxin tolerance are the negative regulatory factor of TLR4 signaling pathway, miRNA, apoptosis, and chromatin modification and gene reprogramming of immune cells.

  • The introduction should contain more information regarding the mechanisms that are activating in the sepsis model such as the immunosuppression. Also the responsible cellular populations for the mediated immunosuppresion are required to be described.

-->We really appreciate your important comments. We added immunosuppression areas into introduction as follows; The mechanisms underlying immunosuppression in sepsis include lymphopenia, immature neutrophil phenotype, loss of monocyte inflammatory cytokine production (endotoxin tolerance), increased myeloid-derived suppressor cells (MDSCs), production of anti-inflammatory cytokines, loss of expression of HLA-DR in antigen presenting cells (dendritic cells and macrophages), programmed cell death 1 receptor and ligand in T cells, increasing immunosuppressive T cell phenotypes such as T helper 2 and T regulatory cells, and apoptosis.

3) The results are well described and no changes are required.

4) Discussion section, lines 137-140, The next generation of treatments could...immune cells. I think that this paragraph cannot be supported by the results of this study. It could be as a conclusion. I would suggest to the authors, that this paragraph needs to be removed.

-->As your recommendations we removed the paragraph in main discussion. However, reviewer 3 recommends that “at least, any treatment options should be suggested based on these in vivo outcomes.” So, we added some comments in limitation section as follows; Also, we should investigate another mechanism about endotoxin tolerance with this model. These might include miRNA and chromatin modification and gene reprogramming of immune cells. With these further studies, we could develop newer treatment options. The next generation of treatments could be separate (or opposite) targets with different treatment options, i.e., anti-inflammatory agents to reduce or antagonize increased plasma cytokines, and immune-enhancing/anti-apoptotic drugs to augment the immune function of immunosuppressed immune cells.

Reviewer 3 Report

 Serial change of endotoxin tolerance in a polymicrobial sepsis model.

 It is known that sepsis-induced immunosuppression remains the leading cause of death in most intensive care units. The immune response that occurs during sepsis is characterized by a cytokine‐mediated hyper‐inflammatory phase, which most patients survive, and a subsequent immunosuppressive phase. Therefore, therapies that improve host immunity might increase the survival of patients with sepsis.

Until recently, most research on sepsis was focused on blocking the initial hyper‐inflammatory response. Initially, the proinflammatory response was believed to be the major cause of mortality in patients with sepsis and was frequently targeted for therapeutic intervention. However, efforts to improve outcomes by targeting proinflammatory cytokines and mediators, such as TNF and IL‐1β antagonists, endotoxin antagonists, Toll‐like receptor (TLR) blockers, and platelet activating factor inhibitors, have been unsuccessful.

Recent studies show that the activation of both proinflammatory and anti‐inflammatory immune responses occurs promptly after the onset of sepsis.

The rapid deaths of patients with sepsis are typically owing to a hyper‐inflammatory “cytokine storm” response. If sepsis persists, the failure of crucial elements of both the innate and the adaptive immune system occurs, such that patients enter a marked immunosuppressive state.

Therefore, it is highly important to know the accurate timing and serial changes in terms of immunosuppression by sepsis to treat the patients properly.

In this regard, this manuscript aimed to demonstrate serial changes in endotoxin tolerance using a polymicrobial sepsis model.

Overall, the concept and approaches are very straightforward, and all experiments seem well-executed, but some questions remain in this version of the manuscript.

 Here are some comments.

 Major comments

1.                  The authors showed the number of inflammatory markers and immune cell markers under sepsis conditions. However, there is no further discussion on how we can translate these results to apply to clinical situations although the reviewer recognizes that it could be highly variable. At least, any treatment options should be suggested based on these in vivo outcomes.

2.                  As the authors mentioned in the discussion section, it is not clear the underlying mechanism of these changes, which is an essential part of this type of study. Therefore, the authors should mention what studies are scheduled and how you will perform the scheduled experiments to understand these mechanisms in the discussion section at least.  

 Minor comments

1.         Many typos and grammatical errors are found throughout the manuscript.

Author Response

Reviewer 3

Serial change of endotoxin tolerance in a polymicrobial sepsis model.

 It is known that sepsis-induced immunosuppression remains the leading cause of death in most intensive care units. The immune response that occurs during sepsis is characterized by a cytokine‐mediated hyper‐inflammatory phase, which most patients survive, and a subsequent immunosuppressive phase. Therefore, therapies that improve host immunity might increase the survival of patients with sepsis.

Until recently, most research on sepsis was focused on blocking the initial hyper‐inflammatory response. Initially, the proinflammatory response was believed to be the major cause of mortality in patients with sepsis and was frequently targeted for therapeutic intervention. However, efforts to improve outcomes by targeting proinflammatory cytokines and mediators, such as TNF and IL‐1β antagonists, endotoxin antagonists, Toll‐like receptor (TLR) blockers, and platelet activating factor inhibitors, have been unsuccessful.

Recent studies show that the activation of both proinflammatory and anti‐inflammatory immune responses occurs promptly after the onset of sepsis.

The rapid deaths of patients with sepsis are typically owing to a hyper‐inflammatory “cytokine storm” response. If sepsis persists, the failure of crucial elements of both the innate and the adaptive immune system occurs, such that patients enter a marked immunosuppressive state.

Therefore, it is highly important to know the accurate timing and serial changes in terms of immunosuppression by sepsis to treat the patients properly.

In this regard, this manuscript aimed to demonstrate serial changes in endotoxin tolerance using a polymicrobial sepsis model.

Overall, the concept and approaches are very straightforward, and all experiments seem well-executed, but some questions remain in this version of the manuscript.

 Here are some comments.

 Major comments

  1. The authors showed the number of inflammatory markers and immune cell markers under sepsis conditions. However, there is no further discussion on how we can translate these results to apply to clinical situations although the reviewer recognizes that it could be highly variable. At least, any treatment options should be suggested based on these in vivo outcomes.

-->We really appreciate your great comments. We need more in-depth experiments to solve the problems. We are investigating various aspects of mechanisms. Simultaneously, we are testing various drugs to target plasma level and immune cell levels. However, these could be included in next papers. We hope you could understand that. Reviewer 2 recommended discussion section, lines 137-140, “The next generation of treatments could...immune cells”. needs to be removed, so we added this to limitation section and describe more suggestions in potential therapeutics as follows; Also, we should investigate another mechanism about endotoxin tolerance with this model. These might include miRNA and chromatin modification and gene reprogramming of immune cells. With these further studies, we could develop newer treatment options. The next generation of treatments could be separate (or opposite) targets with different treatment options, i.e., anti-inflammatory agents to reduce or antagonize increased plasma cytokines, and immune-enhancing/anti-apoptotic drugs to augment the immune function of immunosuppressed immune cells.

  1. As the authors mentioned in the discussion section, it is not clear the underlying mechanism of these changes, which is an essential part of this type of study. Therefore, the authors should mention what studies are scheduled and how you will perform the scheduled experiments to understand these mechanisms in the discussion section at least.

--> We appreciate your great comments. As commented above, we need more in-depth experiments to solve the problems. We are investigating various aspects of mechanisms. Simultaneously, we are testing various drugs to target plasma level and immune cell levels. However, these could be included in next papers. We hope you could understand that. Reviewer 2 recommended discussion section, lines 137-140, “The next generation of treatments could...immune cells”. needs to be removed, so we added this to limitation section as follows; Also, we should investigate another mechanism about endotoxin tolerance with this model. These might include miRNA and chromatin modification and gene reprogramming of immune cells. With these further studies, we could develop newer treatment options. The next generation of treatments could be separate (or opposite) targets with different treatment options, i.e., anti-inflammatory agents to reduce or antagonize increased plasma cytokines, and immune-enhancing/anti-apoptotic drugs to augment the immune function of immunosuppressed immune cells.

 Minor comments

  1. Many typos and grammatical errors are found throughout the manuscript.

-->We did English polishing process through one commercial company before submission, but we found some errors, and made corrections with your recommendations.

Round 2

Reviewer 1 Report

Dear Authors,

thank you for providing comprehensive and convincing answers to my questions and queries and accordingly revised your manuscript.

Best Regards